# miR-150-5p and let-7b-5p in Blood Myeloid Extracellular Vesicles Track Cognitive Symptoms in Patients with Multiple Sclerosis

**DOI:** 10.3390/cells11091551

**Published:** 2022-05-05

**Authors:** Federica Scaroni, Caterina Visconte, Maria Serpente, Maria Teresa Golia, Martina Gabrielli, Marijn Huiskamp, Hanneke E. Hulst, Tiziana Carandini, Milena De Riz, Anna Pietroboni, Emanuela Rotondo, Elio Scarpini, Daniela Galimberti, Charlotte E. Teunissen, Maureen van Dam, Brigit A. de Jong, Chiara Fenoglio, Claudia Verderio

**Affiliations:** 1Institute of Neuroscience, CNR, Via Follereau 3, 20854 Vedano al Lambro, Italy; frc.scaroni@gmail.com (F.S.); mariateresa.golia@in.cnr.it (M.T.G.); martina.gabrielli@in.cnr.it (M.G.); 2Department of Biomedical, Surgical and Dental Sciences, University of Milan, Via F. Sforza 35, 20122 Milan, Italy; caterina.visconte@unimi.it (C.V.); elio.scarpini@policlinico.mi.it (E.S.); daniela.galimberti@policlinico.mi.it (D.G.); 3Centro Dino Ferrari, University of Milan, 20122 Milan, Italy; maria.serpente@policlinico.mi.it (M.S.); tiziana.carandini@policlinico.mi.it (T.C.); milena.deriz@policlinico.mi.it (M.D.R.); anna.pietroboni@policlinico.mi.it (A.P.); emanuela.rotondo@policlinico.mi.it (E.R.); 4Fondazione IRCCS Ca’ Granda, Ospedale Policlinico, 20122 Milan, Italy; 5MS Center Amsterdam, Amsterdam Neuroscience, Department of Anatomy and Neurosciences, Amsterdam UMC, Vrije Universiteit Amsterdam UMC, De Boelelaan 1117, 1081 Amsterdam, The Netherlands; m.huiskamp@amsterdamumc.nl (M.H.); m.vandam2@amsterdamumc.nl (M.v.D.); 6Health-, Medical- and Neuropsychology Unit, Institute of Psychology, Leiden University, 2300 Leiden, The Netherlands; h.e.hulst@fsw.leidenuniv.nl; 7MS Center Amsterdam, Amsterdam Neuroscience, Department of Neurology, Amsterdam UMC, Vrije Universiteit Amsterdam, De Boelelaan 1117, 1081 Amsterdam, The Netherlands; c.teunissen@amsterdamumc.nl (C.E.T.); b.dejong@amsterdamumc.nl (B.A.d.J.); 8Department of Neuropathology and Transplantation, University of Milan, Via F. Sforza 35, 20122 Milan, Italy

**Keywords:** multiple sclerosis, plasma extracellular vesicles, myeloid extracellular vesicles, microRNAs, cognitive deficits, biomarkers

## Abstract

Cognitive deficits strongly affect the quality of life of patients with multiple sclerosis (MS). However, no cognitive MS biomarkers are currently available. Extracellular vesicles (EVs) contain markers of parental cells and are able to pass from the brain into blood, representing a source of disease biomarkers. The aim of this study was to investigate whether small non-coding microRNAs (miRNAs) targeting synaptic genes and packaged in plasma EVs may reflect cognitive deficits in MS patients. Total EVs were precipitated by Exoquick from the plasma of twenty-six cognitively preserved (CP) and twenty-three cognitively impaired (CI) MS patients belonging to two independent cohorts. Myeloid EVs were extracted by affinity capture from total EVs using Isolectin B4 (IB4). Fourteen miRNAs targeting synaptic genes were selected and measured by RT-PCR in both total and myeloid EVs. Myeloid EVs from CI patients expressed higher levels of miR-150-5p and lower levels of let-7b-5p compared to CP patients. Stratification for progressive MS (PMS) and relapsing-remitting MS (RRMS) and correlation with clinical parameters suggested that these alterations might be attributable to cognitive deficits rather than disease progression. This study identifies miR-150-5p and let-7b-5p packaged in blood myeloid EVs as possible biomarkers for cognitive deficits in MS.

## 1. Introduction

Although most work on multiple sclerosis (MS) has been addressing the role of immune cell activation on oligodendrocytes and myelin damage, current knowledge indicates that MS harbors several characteristic aspects of a neurodegenerative disorder, that is, damage to axons, synapses and nerve cell bodies [1,2]. Specifically, synaptic loss and axonal degeneration are early events in the disease [3,4] and could occur independently of demyelination.

Between 40% and 70% of patients suffer from MS-related cognitive impairment, which has been more strongly associated with the neurodegenerative aspects of the disease [5]. Cognitive impairment is present in all MS subtypes and is frequently seen early in the disease course [6,7], representing a predictor of future disease progression [8,9,10]. The cognitive domains most frequently impaired in MS are information-processing speed, working memory [8] and long-term memory [11], followed by dysfunction in executive functioning and attention [12,13]. Mood disturbances and personality abnormalities often occur in MS subjects too, and may also strongly affect patients’ quality of life [9].

Whereas functional deficits in the central nervous system (CNS) are routinely evaluated with the administration of the Expanded Disability Status Scale (EDSS) [14], assessment of cognitive functions by neuropsychological tests is time-consuming, and cognition cannot currently be monitored in standard clinical practice. From this point of view, a biomarker for cognitive deficit would be of great value. Many studies have tried to characterize the structural MRI correlates of cognitive impairment in patients with MS, finding an association between isolated measures of structural CNS damage and cognitive performance [15]. However, isolated MRI measurements, such as thalamic volume loss, are only moderately related to cognitive performance [16,17,18]. Thus, the development of more powerful/complementary cognitive MS biomarkers is still an unmet clinical need.

A full understanding of the molecular mechanisms underlying synaptic changes and altered connectivity in MS could be useful for identifying cognitive biomarkers. Recent evidence has linked synaptic alterations and cognitive deficits in MS to activation of microglia, the immune cells of the brain. Specifically, it was first demonstrated that microglia activation is necessary for hippocampal synaptic and cognitive alterations in experimental autoimmune encephalomyelitis (EAE) [19], an animal model for MS. Activation of NLRP3 inflammasome in microglia was then shown to play an important role in EAE cognitive deficits via the alteration of astrocyte phenotypes and the release of complement component 3 (C3) [20]. Consistently, higher complement deposits were detected in post-mortem hippocampi of MS patients with cognitive impairments, compared to cognitive preserved patients [21]. Other studies linked microgliosis to increased production of microglial extracellular vesicles (EVs) in MS patients and revealed that microglial EVs impair stability of excitatory synapses [22,23].

EVs recently entered the scene of neuroscience as important mediators of cell-to-cell communication, which transfer a variety of biomolecules, including genetic materials, between cells. EVs can pass from the brain into blood and circulate in body fluids, where they are emerging as promising markers for several neurological diseases, representing a brain “fluid biopsy” [24,25]. Among EV cargo which influence excitatory synapses, our recent data point to small non-coding microRNAs (miRNAs), which have validated synaptic targets [23]. By silencing synaptic proteins, these miRNAs, which are delivered to neurons by EVs produced from reactive microglia and astrocytes, impair synapse stability and function in the mouse brain, and may be responsible for defects in synaptic plasticity and cognition in MS patients. In this study, we tested the hypothesis that miRNAs targeting synaptic proteins in myeloid EVs could represent cognitive biomarkers in MS.

## 2. Materials and Methods

### 2.1. Patients Enrolled and Study Design

Patients with MS were included in the study with the following eligibility criteria: (i) diagnosis of MS according to the 2010 McDonald criteria [26], (ii) EDSS score ≤ 7, (iii) age ≥ 25 and ≤70 years, (iv) no steroid treatment for at least four weeks before the blood withdrawal and (v) ability to provide written informed consent. Details of demographic characteristics are shown in Table 1. A first Italian MS cohort from IRCCS Fondazione Ca’ Granda Ospedale Maggiore Policlinico, consisting of 21 patients, was initially enrolled. Patients were grouped as cognitively preserved (CP) and cognitively impaired (CI) according to their neuropsychological status assessed with the Rao Brief Repeatable Battery (BRB) [27] and were characterized for the neurofilament serum levels (sNf-L). Subsequently, a validation Amsterdam MS Center cohort from Amsterdam UMC, location VUmc, was enrolled, consisting of 28 patients. A test battery based on the MACFIMS [28] was administered to the second cohort, as outlined below.

IRCCS Fondazione Ca’ Granda cohort. Eleven CP (all RRMS type) and 10 CI (six RRMS and four PMS) MS patients were recruited at the Neurology Unit of the Fondazione IRCCS Ca’ Granda Ospedale Maggiore Policlinico (Milan, Italy) (Appendix A). Cognitive functions were assessed by an experienced neuropsychologist with the use of the Italian version of the BRB [27]. It includes tests of: (1) attention and speed in information processing (paced auditory serial addition test (PASAT 3 and PASAT 2) and Symbol Digit Modality Test (SDMT)), (2) verbal memory (selective reminding test, long-term storage (SRT LTS), consistent long-term retrieval (SRT CLTR), and delayed (STR D)), and (3) spatial memory (spatial recall test immediate and delayed (SPART and SPART D)). Consistent with previous works [27], those patients who failed at least three tests were considered cognitively impaired, and those who failed less than three tests were considered cognitively preserved.

MS Center Amsterdam cohort. Fifteen CP (fourteen RRMS and one PMS) and thirteen CI (nine RRMS and four PMS) MS patients were recruited at Amsterdam UMC (Amsterdam, the Netherlands) (Appendix A). Cognitive functions were assessed by a test battery based on the MACFIMS [28], and consisted of the following five (sub-)domains: (1) verbal memory (Dutch version of the California Verbal Learning Test version 2), (2) visuospatial memory (Brief Visuospatial Memory Test–Revised), (3) processing speed (SDMT and Stroop Color–Word Test cards I and II), (4) executive function—verbal fluency (Controlled Oral Word Association Test) and (5) executive function—response inhibition (Stroop Color–Word Test interference score). Scores were corrected for age, education and sex when applicable, and transformed into five (sub-)domain-specific z-scores, as well as one composite score for overall cognitive functioning (i.e., average of all z-scores) based on a normative sample of healthy controls. Patients were classified as CI (i.e., ≥1.5 standard deviations (SDs) below the means of healthy controls on ≥20% of the neuropsychological test scores, corresponding to ≥3/11 test scores) or CP (i.e., remainder) [29]. Additional data included disease duration in years, disease type at the time of the visit and EDSS score.

### 2.2. Serum Nf-L Determination

Nf-L levels were determined in serum obtained from 7 mL of blood withdrawal after centrifugation for 10 min at 1500× *g*. sNf-L levels were measured by using an MSD (R-PLEX Human Neurofilament L Kit, according to the manufacturer’s instructions, Meso Scale Discovery, Rockville, MD, USA) in the Italian cohort. Analysis was performed using a QuickPlex SQ 120 instrument (MSD, Rockville, MD, USA) and DISCOVERY WORKBENCH^®^4.0 software. sNf-L levels of the Amsterdam MS Center cohort were measured using a single-molecule array (Simoa) assay to measure the Nf-L levels on an HD-X instrument (Quanterix, Billerica, MA, USA) following the manufacturer’s instructions.

### 2.3. Total EVs and Myeloid EV Isolation

For total and myeloid EVs’ isolation, 7–10 mL of venous blood was drawn by syringe into 0.5 mL of saline with EDTA, centrifuged for 15 min at 1500× *g*, aliquoted and stored at −80 °C pending use.

Aliquots of 500 μL of plasma were incubated with 5 μL of Thrombin (Thrombin Plasma Prep 611 U/mL, System biosciences, SBI, Palo Alto, CA, USA) for 5 min in order to eliminate the large amount of fibrin present in plasma. After centrifugation for 5 min at 3000× *g* at RT, total EVs were harvested from the supernatants by precipitation with 126 µL of Exoquick exosome precipitation solution (EXOQ; System biosciences, SBI, Palo Alto, CA, USA) per 500 μL of supernatants and, after incubation for 30 min at 4 °C, centrifuged for 30 min at RT. Each pellet containing EVs was resuspended in 500 μL of phosphate buffer. To enrich in myeloid EVs, 100 µL aliquots of the total EV suspensions were incubated with 40 μL of streptavidin magnetic beads (Exo-Flow streptavidin 9.1 μm beads, System biosciences, SBI, Palo Alto, CA, USA), preincubated with 10 μL of biotinylated Isolectin B4 (IB4; 1 mg/mL; Sigma Aldrich, St. Louis, MO, USA) at 4 °C with continuous mixing overnight. Then, IB4-positive EVs were isolated using a magnetic plate and treated for subsequent RNA extraction and Western blot analysis. Biotinylated IB4 pretreated with the IB4 interacting molecule Melibiose (6-*O*-a-d-galactopyranosyl-d-glucose, 1M) [22] was used as a negative control for Western blot analysis. For size distribution analysis, myeloid EVs were resuspended in 300 μL of Exosome Elution Buffer (Exo-Flow, System biosciences, SBI, Palo Alto, CA, USA) and incubated on a rotating rank for 40 min at RT to detach IB4-positive EVs from bead–antibody complexes. Final suspensions containing EVs were stored at −80 °C for further analysis. 

### 2.4. Analysis of EV Size and Concentration 

The size distribution and concentration of total and myeloid EVs were measured by the Tunable Resistive Pulse Sensing (TRPS) technique, using the qNano instrument (Izon, Christchurch, New Zealand). Then, 0.1 µm of filtered PBS supplemented with 0.3% Wetting Solution (from Izon EV reagent kit) was used for both pre-treating the pore and suspending EVs in order to prevent EVs binding to the pore or spontaneous EV aggregation. Total and myeloid-enriched EV samples were analyzed using a NP200 nanopore (Izon, Christchurch, New Zealand). CPC200 calibration particles (carboxylated polystyrene particles, supplied by Izon and diluted following the manufacturer’s instructions) were used as standards, and they were measured immediately before or after the experimental samples under identical conditions. Data acquisition and analysis were performed using Izon Control Suite software (version V3.2, Izon, Christchurch, New Zealand).

### 2.5. Western Blotting

The protein concentration of total EVs and IB4-positive EVs isolated with Exo-Flow 2.0 streptavidin beads (System biosciences, SBI, Palo Alto, CA, USA) was quantified by the Micro BCA Protein Assay Kit (Thermo Fisher Scientific, Waltham, MA, USA) to load equal amounts of total EVs and IB4-positive EVs. The negative control (EVs captured by beads coated with melibiose-treated IB4) was immune-captured from an equal volume of total EVs compared to IB4-positive EVs. EVs were dissolved in PBS with LDS sample buffer (Life Technologies, Waltham, MA, USA) and separated using 12% SDS polyacrylamide gels. Then, samples were electro-transferred onto nitrocellulose membranes and blot was incubated overnight at 4 °C with specific primary antibodies: anti-AnnexinA2 (ANXA2) (diluted 1:500; Abcam, Cambridge, MA, USA), anti-TMEM119 (diluted 1:1000; BioLegend, San Diego, CA, USA) and anti-CD171 (diluted 1:1000; Thermo Fisher Scientific, Waltham, MA, USA). After washing, membranes were incubated with horseradish peroxidase-conjugated secondary antibodies (Pierce, Thermo Fisher Scientific, Waltham, MA, USA) for 1 h and immuno-positive bands were detected by enhanced chemiluminescence (ECL, Euroclone, Milan, Italy, or FEMTO, Thermo Fisher, Waltham, MA, USA) according to the manufacturer’s instructions. Densitometric analysis of immunoreactive bands was performed using ImageJ software (www.imagej.nih.gov/ij/, accessed on 1 July 2021).

### 2.6. Selection of miRNAs with Synaptic Targets 

Only a fraction of miRNAs are sorted from cells into EVs [23,30] and are detectable in EVs circulating in human blood [31,32]. In addition, the limited volume of plasma available from biobanks (about 500 µL) prevents multiple analyses on the same patient. Therefore, to increase the chance to identify vesicular miRNAs reflecting/predicting cognitive symptoms in MS patients, we restricted the analysis to a small group of miRNAs with validated synaptic targets in humans. We focused on miRNAs highly expressed in human circulating EVs (top 10 miRNAs enriched in plasma EVs) [32], and/or consistently detected in plasma EVs from healthy subjects/Alzheimer’s disease patients (n = 58) [31] (Appendix A). According to the microRNA database, miRTarBase (http://miRTarBase.mbc.nctu.edu.tw/, accessed on 1 July 2021), and Panther classification (http://pantherdb.org/news/news20201218.jsp, accessed on 1 July 2021), the vast majority of these miRNAs have validated synaptic targets, as listed in Appendix A. To narrow their number, the miRNAs were further selected using the following criteria: enrichment in microglia/peripheral immune cells and EVs thereof [33,34,35,36,37,38,39,40,41,42,43], release under inflammatory or neurodegenerative stimuli [23], previous identification as candidate biomarkers for MS pathology [44,45,46,47,48,49,50,51,52,53,54,55] and dysregulation in neurodegenerative diseases [31,56,57,58]. The miRNAs that passed the selection criteria were the following: miR-146a-5p, miR-223-3p, miR-16-5p, miR-23a-3p, miR-20a, miR-21-5p, miR-125a-5p, miR-146b-5p, miR-150-5p, miR-24-3p, miR-15a-5p, let-7b-5p, miR-451a and miR-126-5p (Table 2). Experimentally validated synaptic targets with strong evidence in humans (by report assays, qPCR and Western blot analysis) of candidate miRNAs are listed in Appendix A. In addition, cel-miR-39, a *C. elegans* miRNA, not expressed in human cells, was selected as an exogenous control for RNA extraction and the retro-transcription process.

### 2.7. Total RNA Isolation and Quantitative Real-Time PCR Analysis

RNA was extracted from total and IB4-positive EVs using the Total Exosome RNA Isolation Kit (Invitrogen, Waltham, MA, USA) following the manufacturer’s protocol. The reverse transcription (RT) reaction was performed using the MicroRNA Reverse Transcription Kit (Applied Biosystem, Waltham, MA, USA). Given the low RNA amount in our samples, cDNA was pre-amplified using TaqMan^®^Pre-Amp Master Mix (Applied Biosystem, Waltham, MA, USA). Each well of a 96-well plate was loaded with 5 μL of pre-amplified cDNA (diluted 1:5) and 15 μL of PCR Reaction Mix, composed of TaqMan^®^ Universal PCR Master Mix, TaqMan^®^ miRNA Assay and RNase- free water (Applied Biosystem, Waltham, MA, USA). The reaction and amplification protocol was performed according to the manufacturers’ instructions. Finally, the quantitative RT-PCR was run on a QuantStudio™ 12K Flex Real-Time PCR System (Thermo Fischer Scientific, Waltham, MA, USA). Raw data underwent quality control analysis, by applying a Crt (relative threshold cycle) cutoff < 28 and the amplification score > 1 (measure of good amplification quality). Candidate miRNAs were detectable in total EVs from most MS patients, except for miR-146a-5p and miR-23a-3p, whose expression levels did not reach good quality control parameters and were excluded from the analysis. In most IB4-positive EVs, miR-15a-5p and miR-125a-5p were undetectable. As an endogenous control reference, miR-16-5p and miR-21-5p were used for the Italian population, while miR-16-5p and miR-451a for the Amsterdam MS Center one, identified through the algorithm GeNorm. The relative quantification (RQ) was calculated with the 2^−ΔΔCt^ method [59]. Patients’ classification in terms of cognitive performance was revealed upon completion of the PCR analysis.

### 2.8. Statistical Analysis

Principal component analysis (PCA) for both cohorts was performed using ClustVis (https://biit.cs.ut.ee/clustvis/, accessed on 16 April 2022). Unit variance row scaling was applied for PCA. Statistical analysis was performed using Prism GraphPad 9.0 software (San Diego, CA, USA). Data distribution was tested for normality with the Kolmogorov–Smirnov and Shapiro–Wilk tests. All data are shown as mean ± SEM. Differences between two groups were analyzed with the Mann−Whitney test. Multiple comparisons were performed by one-way ANOVA followed by Tukey’s test, or the Kruskal–Wallis test followed by Dunn’s Multiple Comparison test as post hoc tests. For EV miRNA enrichment analysis, the Wilcoxon matched-pairs signed-rank test was used. Differences were considered significant when *p* < 0.05 and *p* < 0.01, and they are indicated by one or two asterisks, respectively. The Spearman test for correlation between miR-150-5p/let-7b-5p and biochemical, demographic, clinical variables and neuropsychological test scores was also used.

## 3. Results

We selected 14 miRNAs with validated synaptic targets in humans (Figure 1; for selection criteria see Section 2) and probed these miRNAs for both total and myeloid EVs isolated from plasma of CP and CI MS patients belonging to 2 independent cohorts. Total EVs were isolated from human plasma with the Exoquick commercial kit, and myeloid EVs were extracted from total EVs by immune-affinity capture using the myeloid marker IB4.

### 3.1. Isolation of Total and Myeloid EVs from Blood Plasma

According to MISEV 2018 guidelines [60], we previously characterized, via multiple approaches, i.e., electron microscopy, flow cytometry, Nanoparticle Tracking Analysis (NTA) and Western blotting, human plasma EVs isolated by Exoquick, and a subpopulation affinity-captured from total plasma EVs, finding pure populations of mainly oval-shaped particles, approximately 150 nm in diameter [31]. The presence of miRNAs packaged inside plasma EVs was also previously assessed using an RNase protection assay [31]. In this study, total EVs precipitated with Exoquick from frozen MS plasma samples were further analyzed by TRPS (number of MS patients = 4). The analysis showed that an average of 3.49 × 10^11^ ± 2.62 × 10^11^ EVs can be extracted from 1 mL of plasma in the Exoquick pellet, and confirmed that EVs were relatively small in size, ranging from 84 to 661 nm (mean diameter = 140 nm, mode of the diameter = 110 nm; Figure 2A, left). TRPS analysis showed that myeloid EVs, isolated from total EVs by affinity capture with IB4, were similar in size to total EVs (mean diameter = 144.83 nm, mode of the diameter = 111.83 nm; Figure 2A, right), and were about 1% of the total EVs. 

### 3.2. Assessment of Myeloid EV Enrichment

The specificity of the affinity method to extract myeloid EVs from the total EV population was assessed by Western blotting on total EVs and IB4-positive EVs isolated from a plasma pool of three subjects. Staining for the EV marker ANXA2, the microglia marker TMEM119 and the neural marker CD171 showed immunoreactivity of total EVs for both microglial and neuronal markers (Figure 2B). As expected, densitometric analysis revealed that IB4-positive EVs were enriched in TMEM119 (by 1.75-fold) compared to total EVs, while they were almost negative (depleted to 0.09-fold) for the neuronal marker CD171 (Figure 2B). Less EVs were isolated upon pretreatment of IB4-coated beads with melibiose [22], which inhibits IB4 binding to myeloid cell-specific glucose residues, as evidenced by decreased ANXA2 immunoreactivity (Figure 2C).

### 3.3. Expression of Selected miRNAs in Total and Myeloid EVs

We initially investigated a population consisting of 11 CP MS patients and 10 CI MS patients assessed by a short version of BRB, a series of tests previously used for cognitive evaluation in MS [61]. 

Results of the study cohort characteristics are provided in Table 1. We found an upregulation of miR-24-3p, miR-126-5p, miR-223-3p and miR-146b-5p in total EVs from CI compared to CP patients (RQ ± SEM: 18.82 ± 3.93 vs. 5.93 ± 1.70, *p* = 0.003; 2.70 ± 0.44 vs. 1.22 ± 0.16, *p* = 0.005; 13.01 ± 2.45 vs. 3.50 ± 0.89, *p* = 0.001; 5.41 ± 0.89 vs. 2.99 ± 0.92, *p* = 0.04, respectively, Figure 3A), while in IB4-positive EVs, miR-150-5p was upregulated and let-7b-5p was downregulated in patients with cognitive disabilities (RQ ± SEM: 3.29 ± 0.11 vs. 0.60 ± 0.2, *p* = 0.03; 0.323 ± 0.15 vs. 1.39 ± 0.33, *p* = 0.03, respectively, Figure 3B). 

To validate these data, the expression of the 14 miRNAs was analyzed in a second cohort consisting of 15 CP MS patients and 13 CI MS patients, who underwent a larger battery of neuropsychological tests (7 tests) to deeply measure the most affected cognitive domains in MS. No significant changes in the content of any miRNAs with validated synaptic targets were detected between CI and CP patients in total EVs (Figure 4A). Conversely, analysis of the miRNA cargo confirmed the upregulation of miR-150-5p, a miRNA involved in the development of myeloid lineages [62] and in MS inflammation, and the downregulation of let-7b-5p, a miRNA controlling immune cell responses [63], in IB4-positive EVs from CI patients compared to CP patients (RQ ± SEM: 4.93 ± 0.99 vs. 2.22 ± 0.84, *p* = 0.03; 0.71 ± 0.27 vs. 3.52 ± 1.03, *p* < 0.05, Figure 4B). To exclude the possibility that miR-150-5p and let7b were randomly selected as statistically different, we performed PCA for myeloid EVs in both cohorts. PCA showed that miRNA measurements in myeloid EVs could distinguish patients based on their cognitive status (Appendix A).

As expected for the two miRNAs highly expressed in myeloid cells [33,34,35,36,39], analysis of let-7b-5p and miR-150-5p expression in IB4-positive EVs compared to total EVs revealed an enrichment for both miRNAs in the IB4-positive fraction, albeit the results reached the significant threshold only for miR-150-5p (*p* < 0.0001; Appendix A). This evidence confirmed the enrichment of myeloid EVs in the IB4-positive fraction. Moreover, stratification of let-7b-5p and miR-150-5p levels according to disease type showed no significant differences between PMS and RRMS in both of the cohorts considered (Appendix A), suggesting that the miRNA dysregulation might be attributable to cognitive deficits rather than MS progression.

### 3.4. Correlations between miRNA Levels and Clinical Data

Finally, we performed correlation analysis between miR-150-5p and let-7b-5p expression levels in IB4-positive EVs and clinical features relevant to MS course (age at blood withdrawal, age at disease onset (AAO), gender, EDSS, disease duration), both in the total group and stratified for cognitive status. Correlations between miRNA levels and the axonal damage biomarker Nf-L in the serum were also carried out. miR-150-5p/let-7b-5p expression in myeloid EVs was not correlated with the clinical data, either when patients were grouped or stratified according to their cognitive status (Appendix A). Conversely, a moderate positive correlation between miR-150-5p levels in IB4-positive EVs and sNf-L was observed both in the overall Italian MS population (r = 0.46, *p* = 0.04) and in the Amsterdam MS Center one (r = 0.47, *p* = 0.04). 

## 4. Discussion

By measuring a small set of miRNAs, which may cause synaptic dysfunction, in plasma EVs from two independent small cohorts of CP and CI MS patients, we found a specific signature of cognitive dysfunction, consisting of high levels of miR-150-5p and low levels of let-7b-5p in IB4-positive EVs, derived from microglia and other phagocytes. No significant changes of these miRNAs were detected between CP and CI MS patients in total plasma EVs, which originate from several cell types, indicating that myeloid EVs are a better source for miRNAs reflecting synaptic alterations in MS compared to total plasma EVs and, likely, circulating miRNAs. Although the battery of cognitive tests used was different, both batteries in the two patient cohorts assessed the cognitive domains most frequently impaired in MS.

Overall, our findings suggest that miRNA dysregulation might be attributable to cognitive deficits rather than MS progression. In fact, no differences in the miRNA levels were observed in IB4-positive EVs from PMS compared to RRMS patients, and no direct correlation was found between miRNAs levels in CI patients and sNf-L serum concentration, a proposed predictor of poor clinical outcome in MS [64], whose role in tracking cognitive deficit in MS is still controversial [65,66,67,68]. However, higher miR150-5p levels directly correlated with sNf-L levels in the overall Italian cohort as well as in the Amsterdam MS Center cohort, implying a role of the miRNA in neurodegenerative process. sNf-L were evaluated in the two populations with two different platforms, whose results have already been demonstrated to be highly correlated [69].

A major limitation of our study is the small sample size. Thus, data should be interpreted with caution. This holds true, especially for PMS patients, whose limited number does not allow us to accurately investigate miRNA expression changes between PMS and RRMS. Another apparent limitation is the heterogeneity of the two cohorts, which differ in terms of disease duration and EDSS. However, we would like to point out that the ability of miR150-5p and let-7b-5p to capture cognitive performance in such heterogeneous MS populations is in favor of their value as cognitive biomarkers in all MS forms.

miR-150-5p is a miRNA enriched in both brain-resident microglia [23,33] and peripheral myeloid cells [34,36], which targets many neuronal genes controlling synaptic stability and functions. Among miR-150-5p targets, ephrin type B receptor2 (EPHB2), Gβ5, a member of the signal-transducing G protein β, and Wnt7b regulate dendrite formation, dendritic spines’ maturation and/or stimulate the formation of excitatory synapses [70], while another target, PAK3, mediates cytoskeletal reorganization in dendritic spines during synaptic plasticity [71]. Importantly, defects in PAK3 as well as in Patched domain-containing protein 1 (PTCHD1), another validated miR150-5p target, are associated with intellectual disability in humans [72,73], and PTCHD1 deficiency causes synaptic and cognitive dysfunctions in mice [74]. This suggests that EVs mediate the transfer of miR-150-5p to neurons, and consequent PAK3/PTCHD1 downregulation may be implicated in cognitive impairments of MS patients. Immunohistochemistry of brain tissue from CP and CI patients will be necessary to test this intriguing hypothesis in future studies.

Let-7b-5p targets several relevant synaptic genes, involved in synaptic transmission and cognition, including synaptotagmin1, a fundamental regulator of calcium-dependent exocytosis, Fragile X-Related Protein2 (FXR2), a RNA-binding protein involved in dendritic maturation [75], and neuroplastin (NPTN), a synaptic-enriched adhesion molecule that acts as a key regulator of synaptic plasticity and function [76]. However, let-7b-5p also silences the expression of multiple inflammatory genes in myeloid cells, including type 1 interferon (IFNB1), interleukin 8 (CXCL8) and signal transducer and activator of transcription 2 (STAT2), thereby strongly influencing the immune cell function. Members of the let-7 family were previously shown to be downregulated in microglia exposed to neurodegenerative stimulus [23], and their reduction might be instrumental in bringing about maladaptive myeloid cell activation in neurodegeneration. According to this hypothesis, here, we showed a significant decrease in the expression of let-7b-5p in myeloid EVs from CI compared to CP MS patients, which may reflect the transcriptional and functional signature of dysfunctional microglia in neurodegenerative conditions, termed disease-associated microglia (DAM) [77] or neurodegenerative microglia [78]. Our data are in perfect agreement with a recent study showing that let-7b-5p levels in the cerebrospinal fluid (CSF) directly correlated with a better cognitive performance of RRMS patients, while they were reduced in progressive MS [49]. The latter finding, along with the present study, suggests that let-7b-5p content in myeloid but not total blood EVs might reflect the miRNA levels in the CSF and help to define the activation state of myeloid cells in the course of the disease.

To conclude, our data suggested that let-7b-5p and miR-150-5p profiling in myeloid EVs may represent a possible useful diagnostic tool to measure cognitive dysfunction in MS, which provides complementary information to MRI indication (e.g., thalamic volume loss). Further research is needed, however, to validate the expression changes of the synaptic and/or myeloid targets controlled by miR-150-5p and let-7b-5p, and to see how well the miRNAs perform as biomarkers of cognitive impairment, also in relation to other proposed biomarkers for MS cognitive impairment, such as CSF Tau [79].

## Figures and Tables

**Figure 1 cells-11-01551-f001:**
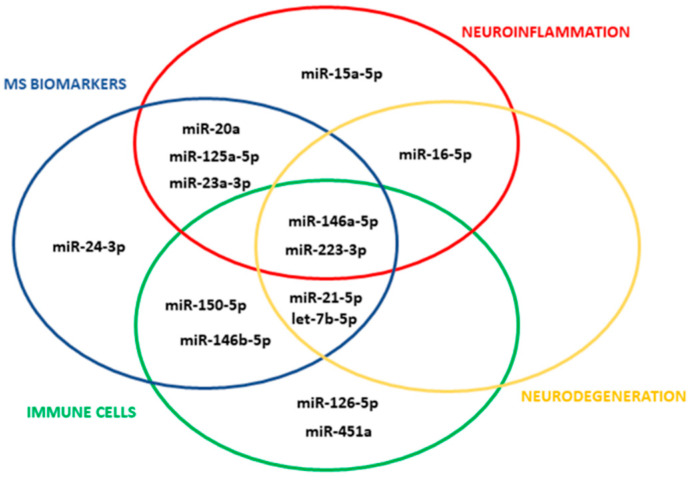
miRNA selection. The Venn diagram shows the 14 circulating miRNAs with validated synaptic targets which meet the selection criteria.

**Figure 2 cells-11-01551-f002:**
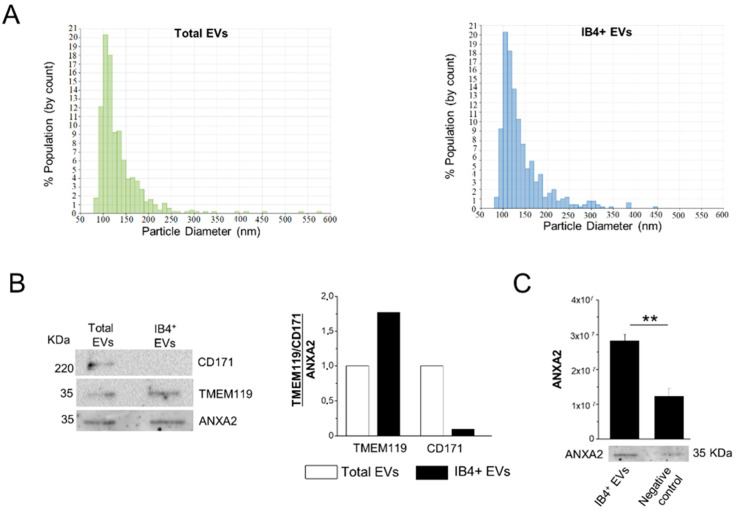
Characterization of total and IB4-positive EVs. (**A**) Representative size distribution of total and IB4-positive (IB4+) EVs detected by TRPS. (**B**) Representative Western blot and relative quantification of microglial TMEM119 marker and neuronal CD171 marker in total EVs and IB4+ EVs. CD171 and TMEM119 expression are normalized to ANXA2 staining. (**C**) Expression and quantification of EV marker ANXA2 in IB4+ EVs and negative control (IB4-coated beads pretreated with melibiose) (n = 3). Data are shown as mean ± SEM. ** *p* < 0.01, one-way ANOVA, Tukey’s multiple comparisons test.

**Figure 3 cells-11-01551-f003:**
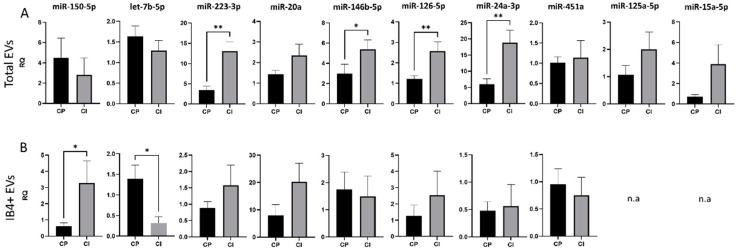
miRNA cargo in total and IB4-positive (IB4+) EVs in the Italian cohort of MS patients. (**A**) RQ of selected miRNA levels in total plasma EVs from CP and CI patients. (**B**) RQ of selected miRNA levels in IB4+ EVs from CP and CI patients. Data were normalized to miR-16-5p and miR-21-5p, expressed as 2^–ΔΔCt^. Mann–Whitney test, * *p* < 0.05, ** *p* < 0.001.

**Figure 4 cells-11-01551-f004:**
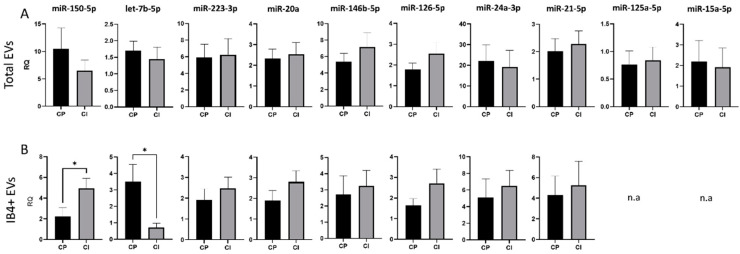
miRNA cargo in total and IB4-positive (IB4+) EVs in the Amsterdam MS Center cohort of MS patients. (**A**) RQ of selected miRNA levels in total plasma EVs from CP and CI patients. (**B**) RQ of selected miRNA levels in IB4+ EVs from CP and CI patients. Data were normalized to miR-16-5p and miR-451a, expressed as 2^–ΔΔCt^. Mann–Whitney test, * *p* < 0.05.

**Table 1 cells-11-01551-t001:** Demographic and clinical features of the two cohorts of patients enrolled.

Italian Cohort				
	Variable	All Patients	RRMS	PMS
	N	21	17	4
	Age, years (range)	42 (21–69)	39.1 (21–65)	54.8 (34–69)
	Age at onset, years (range)	36.2 (20–56)	35.1 (20–56)	46.2 (36–56)
	Gender F:M	12:9	11:6	1:3
	EDSS	2.4 (0–7)	1.8 (0–3.5)	2.4 (5–7)
	Disease duration, years (range)	6.8 (1–15)	5.23 (1–10)	13.5 (12–15)
**Amsterdam Cohort**				
	N	28	23	5
	Age, years (range)	44.9 (29–68)	40.4 (26–66)	57.6 (49–68)
	Age at onset, years (range)	31.5 (20–56)	31 (23–44)	35.7 (18–56)
	Gender F:M	21:7	19:4	2:3
	EDSS	3.8 (2–6.5)	4.75 (2–5.5)	5.1 (3–6.5)
	Disease duration, years (range)	14.9 (1.44–29.3)	12.9 (1.4–31.6)	28.1 (12.4–34.6)

Abbreviations: MS = multiple sclerosis; RRMS = relapsing-remitting MS; PMS = progressive MS; N = number of patients; EDSS = Expanded Disability Status Scale; F = female; M = male.

**Table 2 cells-11-01551-t002:** Characteristics of selected miRNAs targeting synaptic genes.

miRNA	Highly Expressed/Present in Human Serum/Plasma EVs	Enriched in Microglia/Peripheral Myeloid Cells	Present in EVs from Brain/ Peripheral SMyeloid Cells	Putative MS Biomarkers	Dysregulated in Neurodegenerative Diseases
miR-146a-5p	Refs. [31,32]	Refs. [23,33]	Ref. [23]	Refs. [44,46]	Ref. [56]
miR-223-3p	Ref. [31]	Ref. [33]	Ref. [23]	Refs. [44,54]	Ref. [56]
miR-16-5p	Ref. [31]		Ref. [23]		Refs. [31,57]
miR-23a-3p	Ref. [31]		Ref. [23]	Ref. [44]	
miR-20a	Ref. [31]		Ref. [23]	Refs. [52,55]	
miR-21-5p	Ref. [31]	Refs. [37,38]		Refs. [46,51]	Ref. [56]
miR-125a-5p	Refs. [31,32]			Ref. [50]	
miR-146b-5p	Ref. [31]	Ref. [41]		Ref. [46]	
miR-150-5p	Ref. [31]	Refs. [33,34,36]		Ref. [53]	
miR-24-3p	Ref. [31]			Refs. [45,48]	
miR-15a-5p	Ref. [31]		Ref. [23]		
let-7b-5p	Refs. [31,32]	Refs. [35,39]	Ref. [42]	Refs. [47,49,50]	Refs. [42,58]
miR-126-5p	Ref. [31]	Ref. [33]	Ref. [43]		
miR-451a	Refs. [31,32]	Refs. [36,40]	Ref. [43]		

## Data Availability

The data that support the findings of this study are available from the corresponding author upon reasonable request.

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
