# Peer review of "miR-150-5p and let-7b-5p in Blood Myeloid Extracellular Vesicles Track Cognitive Symptoms in Patients with Multiple Sclerosis"

_cells, 2022, doi:10.3390/cells11091551_

Round 1

Reviewer 1 Report

The authors have made a substantial addition required for the acceptance of their manuscript.

Author Response

We thank the Referee for the positive evaluation of our study.

Reviewer 2 Report

This study is well-designed, and the results will contribute to our understanding of the cognitive impairment of MS. I only have comments and suggestions regarding the EV part of the study.

  • The authors didn't provide evidence why EV miRNAs are better than circulating miRNAs to assess cognitive impairment in MS. 
  • I suggest that the authors refer to MISEV 2018 guidelines for the characterization of EVs. The authors should present better WB images.
  • The authors have cited their previous work for the EV characterization, which provided enough details. However, the previous study is not focused on IB4+ EV characterization. Therefore, the authors should provide more evidence for the enrichment of IB4+ EVs. 
  • Densitometric analysis lacks TMEM119 WB results and makes it hard to interpret IB4+ EVs enriched in TMEM119 marker. 
  • Clinical Data correlation results require a table.
  • Does Nf-L measurement with two different platforms be comparable? What is the correlation between Nf-L measurements of two platforms? Normalization of the Nf-L measurements across the cohorts might be helpful to assess better correlation data with EV-associated miRNAs.

Author Response

This manuscript is a resubmission of an earlier submission. The following is a list of the peer review reports and author responses from that submission.

Round 1

Reviewer 1 Report

The authors of the submitted manuscript have screened EV samples from two cohorts of MS patients depending  on their cognitive impairment status. The results of the study highlight two miRs being deregulated between CI and non-CI patients that they could be potentially exploited in further clinical use as a biomarker. The study uses well defined conventional experiments however the overall design of the study needs substantial improvement. Below are the analytical comments:

Study design

-Don't you think that another cohort should be included, namely healthy subjects? This additional cohort would explain maybe the mechanistics behind the role of these miRs in CI in MS. If your conclusions draw on the implication of these miRs in CI, wouldn't for example elevated levels of these miRs between non-CI MS patients and healthy controls provide a different approach in your discussion?

-Can you provide more details on whether the CI patients belong to the RRMS or PMS phenotype?

Technical part

-In the real time pcr section you are stating that results are normalized against miR-16. According to Figure 1, miR-16 belongs to the miRs that are deregulated in neuroinflammation and neurodegeneration processes. Since MS is a predominant neuroinflammatory disease with neurodegenerative processes taking place, as it is stated in the introduction as well, how is it possible to normalize the expression of miR expression against miR-16? Besides that, one reference gene for normalizing Real Time-PCR results is not acceptable. Two are the minimum. 

Statistics

-In the demographics section there are differences between cohorts. For example, the inclusion of genders is not equal between cohorts. Or the mean age is different between cohorts. Is it rational that for PMS the mean age will be higher than RRMS for example? Why haven't you tried to bring the demographic statistics in close proximity? And my overall question is have you checked whether these differences are statistically different, hence maybe affecting the levels of miRs? miRs expression has been shown to be affected from age and gender. Especially Let-7b (Sangiao-Alvarellos et al, Endocrinology, 2013)

-When multiple parameters are analyzed, there is a chance that some parameters will be statistically different due to the vast amount of the parameters. Have you performed multiple comparison methods to exclude the possibility that these two microRNAs are randomly picked up as being statistical different?

Discussion

-In the discussion section you are listing the implication of let-7b and miR-150 in neurodegenerative processes. Could you provide possible mechanistic links between the deregulated miRs and potential mRNA targets that may affect the processes involved in CI? How are these miRs being expressed in EVs affect the disease progression?

Author Response

Point 1: Don't you think that another cohort should be included, namely healthy subjects? This additional cohort would explain maybe the mechanistics behind the role of these miRs in CI in MS. If your conclusions draw on the implication of these miRs in CI, wouldn't for example elevated levels of these miRs between non-CI MS patients and healthy controls provide a different approach in your discussion?

 Response 1: In response to the Reviewer’s comment, we would like to point out that the aim of our study is to look for a cognitive biomarker in MS. The hypothesis driving this study is that under chronic non-resolving inflammation, microglia/macrophages release EVs enriched in miRNAs targeting synaptic genes, which, upon transfer to neurons (ref 23), cause synaptic alterations and cognitive dysfunction. That’s why, as described in the Methods (paragraph 2.6) we first selected miRNAs highly enriched in plasma EVs that target synaptic genes and then focused on those upregulated in microglia/macrophages (and their EVs) under neuro-inflammatory/degenerative conditions, especially those previously identified as candidate MS biomarkers and/or dysregulated in neurodegenerative diseases.

miR-150-5p was previously detected at higher levels in the CSF of a large cohort of MS patients compared to healthy subjects. Immune cells were suggested as the source of miR-150-5p, and the miRNA levels were suggested to reflect immune activation (ref 53). Thus, it is possible that miR-150-5p content is also higher in myeloid EVs from CP patients compared to healthy controls. However, the higher the miRNAs content in myeloid EVs the higher microglial response and the likelihood that cognitive functions are impaired in MS patients due to microglia-neuron miRNA transfer and synaptic gene silencing.

Point 2: Can you provide more details on whether the CI patients belong to the RRMS or PMS phenotype?

Response 2: A new table (Supplementary Table S1) was added with the description of the gender and MS type distribution. Moreover, some lines have been added in the text (pag. 3 - paragraph 2.1) to specify the patients’ MS form: “Eleven CP (all RRMS type) and 10 CI (6 RRMS and 4 PMS) MS patients were recruited at the Neurology Unit of the Fondazione IRCCS Ca' Granda Ospedale Maggiore Policlinico (Milan, Italy) (Supplementary Table S1)”. “Fifteen CP (14 RRMS and 1 PMS) patients and 13 CI (9 RRMS and 4 PMS) MS patients were recruited at Amsterdam UMC (Amsterdam, the Netherlands) (Supplementary Table S1)”.

Point 3: In the real time pcr section you are stating that results are normalized against miR-16. According to Figure 1, miR-16 belongs to the miRs that are deregulated in neuroinflammation and neurodegeneration processes. Since MS is a predominant neuroinflammatory disease with neurodegenerative processes taking place, as it is stated in the introduction as well, how is it possible to normalize the expression of miR expression against miR-16? Besides that, one reference gene for normalizing Real Time-PCR results is not acceptable. Two are the minimum.

Response 3: Despite previous studies indicated that miR-16 is altered in MS patients compared to healthy subject, here we are comparing miRNA expression between two categories of MS subjects, i.e. CP and CI MS patients, where the miR-16  is stable, as indicated by the algorithm GeNorm. Accordingly to Mestdagh et al. 2009, the algorithm GeNorm can be used to identify the most stable endogenous miRNA in small-scale experiments where a small set of miRNAs are analyzed. Therefore, we respectfully think that miR-16 can be used for miRNA normalization.

We agree with the Reviewer that it would be better to normalize data against a second miRNA. Thus, to point out this limitation of our study we have now introduced in the revised text (pag.12) the following sentences: “A technical limitation is due to the use of one single miRNA, miR-16-5p, as endogenous control, although several studies indicate miR-16-5p as a good endogenous control in EV miRNA expression profiling (Refs. 68,69)” (line 399-401).

Point 4: In the demographics section there are differences between cohorts. For example, the inclusion of genders is not equal between cohorts. Or the mean age is different between cohorts. Is it rational that for PMS the mean age will be higher than RRMS for example?  Why haven't you tried to bring the demographic statistics in close proximity? And my overall question is have you checked whether these differences are statistically different, hence maybe affecting the levels of miRs? miRs expression has been shown to be affected from age and gender. Especially Let-7b (Sangiao-Alvarellos et al, Endocrinology, 2013).

Response 4: We thank the reviewer for these considerations and questions that give us the possibility to better explain the differences between cohorts.

-The inclusion gender is unbalanced between female and male as it reflects the gender distribution naturally occurring in the disease. Multiple Sclerosis is more frequent in women than men with a 2:1 ratio. This difference is also reflected in the sample cohort distribution.

- In patients with progressive MS, disease duration is usually longer and mean age higher (Compston and Compston, Lancet Neurology 2008), as occurring in the cohort considered in the present study.

-Clinical variables' influence on miRNAs levels was tested. We found no influence of age at disease onset (AAO), disease duration, and gender on miRNA levels. This piece of information is now better explained in the revised text (pag.11 – paragraph 3.4) by the addition of the following sentences: “miR-150-5p/let-7b-5p expression in myeloid EVs was not correlated with the clinical data, either when patients were grouped or stratified according to their cognitive status”.

Point 5: When multiple parameters are analyzed, there is a chance that some parameters will be statistically different due to the vast amount of the parameters. Have you performed multiple comparison methods to exclude the possibility that these two microRNAs are randomly picked up as being statistical different?

Response 5: We thank the Reviewer for this comment. To exclude the possibility that miR-150-5p and let7b were randomly picked up as being statistical different, we performed a Principal Component Analysis (PCA) for myeloid EVs in both cohorts. Although the number of variables analysed is quite small, the PCA showed a separation of patients. This suggests that miRNA expression data are able to discriminate patients based on cognitive performance. PCA has been described in the Methods (paragraph 2.8). On page 7 we report: “Principal component analysis (PCA) for both cohorts was performed using ClustVis (https://biit.cs.ut.ee/clustvis/). Unit variance row scaling was applied for PCA”. Data from PCA analysis have been included as supplementary figures and described in the Result (pag.10) as follows: “To exclude the possibility that miR-150-5p and let7b were randomly selected as statistically different, we performed PCA for myeloid EVs in both cohorts. PCA showed that miRNA measurements in myeloid EVs could distinguish patients based on their cognitive status (Supplementary figure S1)”. 

Point 6: In the discussion section you are listing the implication of let-7b and miR-150 in neurodegenerative processes. Could you provide possible mechanistic links between the deregulated miRs and potential mRNA targets that may affect the processes involved in CI? How are these miRs being expressed in EVs affect the disease progression?

Response 6: We thank the Reviewer for this comment that prompted us to clarify in the introduction (lines 89-90) that EVs released by microglia were previously reported to transfer their miRNA cargo to neurons, where they silence their target genes (ref 23). Based on these previous findings, we hypothesize that transfer of miR150-5p from myeloid cells to neurons via EVs may downregulate miR150-5p synaptic targets in neurons. In the revised discussion (line 415-416) we now suggest that: “EVs-mediate transfer of miR-150-5p to neurons and consequent PAK3/PTCHD1 downregulation may be implicated in cognitive impairments of MS patients”

Note: the text in yellow has been introduced in the revised manuscript.

Reviewer 2 Report

The study used multiple sclerosis (MS) patient-derived blood myeloid extracellular vesicles (EVs) for isolation of microRNAs (miRNAs) and identification of specific miRNAs as biomarkers for cognitive deficits in MS. The study is interesting. However, following things are concerning and need to be addressed for major improvement of the manuscript.

(1) Small sample size is a limitation, which authors have acknowledged in the Discussion but have not used a sensible solution. Based on this, data need to be interpreted carefully and conclusion needs to be made with reservation.

(2) Higher miR-150-5p levels correlated with serum neurofilament light chain (sNf-L) levels positively in two cohorts but negatively in another cohort, indicating that data needed to be interpreted cautiously to link high miR-150-5p level with cognitive deficits in MS patients.

(3) Western blots shown in Figure 2 are of poor quality. Loading control needs to be uniformly expressed and clearly mentioned. 

(4) In Figure 3B, authors showed increased miR-150-5p and decreased let-7b-5p in IB4 positive EVs to conclude cognitive deficits in MS patients, without validating any of their targets. Suppl Tables 1 and 2 show the possible targets. But actual data from the MS patients were not presented. As such, this study remains descriptive without mechanistic insights.

(5) sNf-L is an established biomarker for neuro-axonal damage in MS patients. Authors did determine the sNf-L levels in three cohorts. An increase in sNf-L shows neuronal damage definitely and cognitive deficits hardly. This study at this stage lacks strong correlation between levels of the identified miRNAs and cognitive deficits in MS patients. It is a study of miRNAs for tracking axonal damage (not cognitive symptoms) in MS patients.

(6) There are no well-characterized biomarkers for cognitive impairment in MS patients. It is not sNF-L but something else more convincing is needed for correlating miRNA levels with cognitively impaired MS patients in this study. Probably, Tau level in cerebrospinal fluid would serve as a better indicator for cognitive impairment (Virgilio E et al., J Neurol 2022).

Author Response

Point 1: Small sample size is a limitation, which authors have acknowledged in the Discussion but have not used a sensible solution. Based on this, data need to be interpreted carefully and conclusion needs to be made with reservation.

 Response 1: We definitely agree with the Reviewer that the small sample size is a limitation of our study and that data should be interpreted cautiously. Nevertheless, having found the same changes in miR-150-5p and let-7b content in myeloid EVs in two independent MS cohorts, albeit of small size, we are confident about the reliability of our results. In response to the Reviewer’s comment, we have now better highlighted this limitation of our study in the discussion (pag.12) as follows: “A major limitation of our study is the small sample size. Thus, data should be interpreted with caution. This holds true, especially for PMS patients, whose limited number does not allow us to accurately investigate miRNA expression changes between PMS and RRMS” (line 396-398).

Point 2: Higher miR-150-5p levels correlated with serum neurofilament light chain (sNf-L) levels positively in two cohorts but negatively in another cohort, indicating that data needed to be interpreted cautiously to link high miR-150-5p level with cognitive deficits in MS patients.

Response 2: We apologise with the Reviewer for not having presented results in a sufficiently clear way. Actually, miR-150-5p levels in myeloid EVs positively correlate with sNf-L in both the Italian and Amsterdam cohort. Thus, the result is concordant in the two independent MS populations that we have analyzed. The negative correlation was found between miR-150-5p levels and age at disease onset (AOO) in the Amsterdam population. To increase the clarity of the description, we revised paragraph 3.4 as follows: “Finally, we performed correlation analysis between miR-150-5p and let-7b-5p expression levels in IB4 positive EVs and clinical features relevant to MS course (age at blood withdrawal, age at disease onset (AAO), gender, EDSS, disease duration) both in the total group and stratified for cognitive status. Correlations between miRNA levels and the axonal damage biomarker Nf-L in the serum were also carried out. miR-150-5p/ let-7b-5p expression in myeloid EVs was not correlated with the clinical data, either when patients were grouped or stratified according to their cognitive status. Conversely, a positive correlation between miR-150-5p levels in IB4-positive EVs and sNf-L was observed both in the overall Italian MS population (r=0.57, P=0.04) and in the Amsterdam MS one (r=0.77, P=0.02).  Furthermore, a trend to a negative correlation between AAO and let-7b-5p (r=-0.48, P=0.05) was found in the Italian MS population. Similarly, in the total Amsterdam population a negative correlation between AAO and miR-150-5p level in myeloid EVs was observed (P=0.04 r=-0.38).

Point 3: Western blots shown in Figure 2 are of poor quality. Loading control needs to be uniformly expressed and clearly mentioned. 

 Response 3: In response to the Reviewer’s comment, we would like to point out that it is difficult to have the loading control (the EV protein annexin-A2) equally expressed among different populations of EVs, i.e. total EVs versus myeloid EVs. This is because loading EV samples at equal protein concentrations, as specified in methods, do not necessarily result in loading of equal EV numbers, due to the presence of protein contaminants. This explains the slight changes (non-significant) in  annexin-A2 staining between total EVs and myeloid EVs in figure 2B (left). Instead, as expected, annexin-A2 staining is significantly lower in the negative control (EV captured by beads coated with melibiose-treated IB4) compared to IB4-positive EVs (EVs captured by beads coated with IB4).This indicates that treatment with melibiose prevents the capacity of lectin IB4 to bind sugar residues on myeloid EVs. Residual annexin-A2 staining indicates unspecific binding of EVs to the beads used for affinity capture. Note that IB4 positive EVs and negative control were immune captured from equal volumes of total EVs. This has now been made clear in the methods (pag.5 - paragraph 2.5) as follows: “The protein concentration of total EVs and IB4 positive EVs isolated with Exo-Flow 2.0 streptavidin beads (System biosciences, SBI, Palo Alto, CA, USA) was quantified by Micro BCA Protein Assay Kit (ThermoFisher Scientific, Waltham, MA, USA) to load equal amount of total EVs and IB4 positive EVs. The negative control (EVs captured by beads coated with melibiose-treated IB4) was immune captured from an equal volume of total EVs compared to IB4 positive EVs”.

The quality of the blot in figure 2 has now been improved and in corresponding quantification control bars have been included for clarity.

Point 4: In Figure 3B, authors showed increased miR-150-5p and decreased let-7b-5p in IB4 positive EVs to conclude cognitive deficits in MS patients, without validating any of their targets. Suppl Tables 1 and 2 show the possible targets. But actual data from the MS patients were not presented. As such, this study remains descriptive without mechanistic insights.

Response 4: We thank the reviewer for this comment. A sentence has been added to the revised text (pag.13) in order to underlie the importance of future validation of miR-150-5p/let-7b-5p targets in MS patients: “Further research, however, is needed to validate the expression changes of the synaptic and/or myeloid targets controlled by miR-150 and let-7b-5p and to see how well the miRNAs perform as biomarkers of cognitive impairment, also in relation to other candidate biomarkers for MS cognitive impairment, such as CSF Tau (Ref.79) ” (line 442-443).

Point 5: sNf-L is an established biomarker for neuro-axonal damage in MS patients. Authors did determine the sNf-L levels in three cohorts. An increase in sNf-L shows neuronal damage definitely and cognitive deficits hardly. This study at this stage lacks strong correlation between levels of the identified miRNAs and cognitive deficits in MS patients. It is a study of miRNAs for tracking axonal damage (not cognitive symptoms) in MS patients.

Response 5: We agree with the Reviewer about the lack of a reliable biomarker for cognitive impairment in MS. In the present work we tried to identify a new cognitive biomarker in MS among the  miRNAs enriched in myeloid EVs, driven by the hypothesis that under chronic non-resolving inflammation, microglia/macrophages release EVs enriched in miRNAs targeting synaptic genes.  Such miRNAs, upon transfer to neurons, cause synaptic alterations (ref 23) and may therefore contribute to cognitive dysfunction in patients characterized by chronic inflammation. We took advantage of sNfl levels, available at both the Italian and the Amsterdam cohort to investigate a possible correlation between the expression levels of miRNAs dysregulated in CI patients and a proposed predictor of axonal damage/poor clinical outcome. We  are aware that the role of  Nf-L in tracking cognitive deficit in MS is still controversial and tried to discuss this aspect (pag.11) as follows: “No differences in the miRNA levels were observed in IB4 positive EVs from PMS compared to RRMS patients, no direct correlation was found between miRNAs levels in CI patients and sNf-L serum concentration, a proposed predictor of poor clinical outcome in MS [63], whose role in tracking cognitive deficit in MS is still controversial (Refs. 64-67)” (line 385-388).

Point 6: There are no well-characterized biomarkers for cognitive impairment in MS patients. It is not sNF-L but something else more convincing is needed for correlating miRNA levels with cognitively impaired MS patients in this study. Probably, Tau level in cerebrospinal fluid would serve as a better indicator for cognitive impairment (Virgilio E et al., J Neurol 2022).

Response 6: We agree with the reviewer on the possible usefulness of CSF Tau level as an indicator of cognitive impairment. In response to the reviewer’s comment, we have modified the last sentence of the discussion as follows: “Further research, however, is needed to validate the expression changes of the synaptic and/or myeloid targets controlled by miR-150 and let-7b-5p and to see how well the miRNAs perform as biomarkers of cognitive impairment, also in relation to other proposed biomarkers for MS cognitive impairment, such as CSF Tau (Ref.79)” (line 444-445).

Note: the text in yellow has been introduced in the revised manuscript.

Round 2

Reviewer 1 Report

The authors have made important changes. However, including an extra reference gene is mandatory for normalization of your results, since Real Time PCR is a key method for your paper. Not having adjusted for that, you results cannot be taken as granted.

Reviewer 2 Report

No more comments.